# DCT-DiffStride: Differentiable Strides with Real-Valued Data

## Abstract

Reducing the size of intermediate feature maps within various neural network architectures is critical for generalization performance, and memory and computational complexity. Until recently, most methods required downsampling rates (*i.e.*, decimation) to be predefined and static during training, with optimal downsampling rates requiring a vast hyper-parameter search. Recent work has proposed a novel and differentiable method for learning strides named DiffStride which uses the discrete Fourier transform (DFT) to learn strides for decimation. However, in many cases the DFT does not capture signal properties as efficiently as the discrete cosine transform (DCT). Therefore, we propose an alternative method for learning decimation strides, DCT-DiffStride, as well as new regularization methods to reduce model complexity. Our work employs the DCT and its inverse as a low-pass filter in the frequency domain to reduce feature map dimensionality. Leveraging the well-known energy compaction properties of the DCT for natural signals, we evaluate DCT-DiffStride with its competitors on image and audio datasets demonstrating a favorable tradeoff in model performance and model complexity compared to competing methods. Additionally, we show DCT-DiffStride and DiffStride can be applied to data outside the natural signal domain, increasing the general applications of such methods.

## 1 Introduction

Dimensionality reduction is a necessary computation in nearly all modern data and signal analysis. In convolutional networks, this reduction is often achieved through strided convolution that decimates a high-resolution input (*e.g.*, images) into a lower-resolution space. Prediction in this lower-resolution space can enable the network to focus on relevant classification features while ignoring irrelevant or redundant features. This reduction is vital to the information distillation pipeline in neural networks, where data is transformed to be increasingly sparse and abstract (Chollet, 2021). On image data, decimation comes in the form of reducing spatial dimensions (*e.g.*, width and height). On time-series data, decimation is performed along the temporal axis.

Decimation can be performed in a variety of ways including statistical aggregation, pooling methods (*e.g.*, max and average pooling), and strided convolutions. By reducing the size of intermediate feature maps, computational and memory complexity of the architecture are reduced. Fewer operations, such as multiplies and accumulates (MACs), are needed to produce feature maps as there are there are fewer values for the network to operate on and store. In convolutional neural networks (CNNs), decimation allows for an increase in the receptive field as subsequent kernels have access to the downsampled values that span several frames. Decimation also allows for increased generalization performance because lower-resolution feature maps allow the network to be resilient to redundancy and ignore spurious activations and outliers that would otherwise not be filtered out.

Most decimation methods, however, still require massive hyper-parameter searches to find the optimal window sizes and strides on which to operate. This puts the onus on post-processing and cross-validation to find optimal values which may not be time or computationally feasible. To address this concern Riad *et al.* (2022) proposed *DiffStride*, which learns a cropping mask in the Fourier domain enabling a differentiable method to find optimal downsampling rates. Thus the decimation is learned through gradient-based methods, rather than requiring massive hyper-parameter searches.

Importantly, (Riad *et al.*, 2022) utilized the discrete Fourier transform (DFT), learning cropping size in the Fourier frequency domain. However, in many applications the DFT can often be outperformed by the discrete cosine transform (DCT) (Ahmed *et al.*, 1974) due to differing periodicity assumptions and better energy compaction. We argue the use of the DCT has several advantages over the DFT, particularly for real-valued data and many natural signals.

We introduce *DCT-DiffStride*, which leverages the advantages of the DCT to learn decimation rates in a CNN. Our contributions in this work are summarized as follows: first, we leverage the improved energy compaction properties of the DCT over the DFT, enabling smaller feature maps in CNNs without substantial loss in model performance. Second, we examine the tradeoff in model complexity and model performance for both DiffStride and DCT-DiffStride across a range of datasets including audio, images, and communications signals. In all tested situations, we conclude that DCT-DiffStride is superior or comparable to DiffStride. Third, we show that these methods can be applied outside the natural signal domain even though the motivation behind using the DCT/DFT as dimensionality reduction techniques is founded in natural signals (the use of low-pass filters), increasing the potential applications of such methods.

While better performance at lower model complexities is useful in many cases, it is also important that energy components are highly concentrated in the lower frequencies for the implementations of DCT-DiffStride and DiffStride. Because the learned strides are cutoff frequencies creating a low-pass filter, we hypothesize that there is less energy being cropped from the signal in the higher frequencies and thus lower information loss. A property of many signals is that energy is not spread uniformly throughout the spectrum but is highly concentrated in the lower frequencies. For many data sources, including natural signals, this property could make the DCT preferable to the DFT.

## 2 FREQUENCY-BASED ANALYSIS

Naturally occurring signals are often defined as signals that humans have evolved to recognize like speech (Singh & Theunissen, 2003) and natural landscapes (Torralba & Oliva, 2003; Ruderman, 1994). These natural signals typically have more low frequency content than high frequency content—that is, signals are typically smooth and change gradually. Taking advantage of this observation, Rippel *et al.* (2015) implement low-pass filters in the Fourier domain enabling fractional decimation rates, which helps to improve the pooling operation and ensure resolution is not reduced too early in the network. This allows for most of the content of the signal to remain present in subsequent layers while still providing a method to reduce dimensionality.

Building on the idea of utilizing Fourier domain learning, Pratt *et al.* (2017) introduced Fourier or Spectral Convolutional Networks, a method for optimizing filter weights using the DFT. This approach optimized convolutional filters in the Fourier domain without conversion to spatial representations. However, without an efficient decimation scheme, these networks exploded in the number of trainable parameters. Lin *et al.* (2019) leveraged the Fourier Convolution Theorem to increase the computational efficiency of pre-trained CNNs, adapting convolutional operations to use the DFT. Chi *et al.* (2020) expanded this idea to cross scale DFTs that employ Fourier units in multiple branches.

Wood & Larson (2021) used learned functions in the Fourier domain to filter signals dynamically. The parametric spectral functions often worked to preserve low frequency content, although a specific cropping mechanism was not employed; therefore, reduction in computational complexity was not investigated. Due to the concentration of information in low frequency content, Rippel *et al.* (2015) introduced *spectral pooling* enabling fractional decimation rates and mitigates information loss compared to other pooling methods like max-pooling. A fractional decimation methodology was similarly proposed using Winograd algorithms for acceleration (Pan & Chen, 2021). These important works led to the innovations in DiffStride Riad *et al.* (2022) and our proposed work for DCT-DiffStride.

In signal analysis and signal classification, it is often desirable to use a linear transformation that tends to compact a large fraction of a signal's energy into just a few transform (or "spectral") coefficients. Let us first define an $N$-dimensional real-valued discrete signal as components of the vector, $\mathbf{x} \in \mathbb{R}^N$. The optimal linear transformation matrix, $\mathbf{T}$, for energy compaction is comprised of column vectors that are the eigenvectors of the covariance matrix of $\mathbf{x}$ with itself, which is the

autocorrelation matrix, $\mathbf{R_x} = Cov\{\mathbf{xx}\} = E\{\mathbf{xx}^T\}$. This result has been known since at least the 1933 publication by Hotelling (1933) and $\mathbf{T}$ is commonly referred to as the Karhunen-Loève Transform (KLT) (Loève, 1945; Karhunen, 1946).

In terms of comprising independent energy components, the DFT is often used due to the Wiener-Khinchine theorem (Ziemer & Trantor, 1985) that states the power spectral density of a physical signal is equivalent to the Fourier transform of the signal's autocorrelation function. The DFT is optimal whenever $\mathbf{x}$ is comprised of periodic signals. Thus, the DFT is equivalent to the KLT for a periodic signal because it is the optimal transformation matrix with respect to energy compaction (Pearl, 1973). As a signal loses its periodic structure, the DFT becomes less optimal in terms of energy compaction because additional basis vectors have significant energies when representing $\mathbf{x}$. This introduces limitations on the rate of decimation for non-periodic signals. Additionally, the spectral coefficients comprising the DFT transform are complex, even when the transformed signal $\mathbf{x}$ is real-valued. These properties of the DFT can be problematic when incorporated with CNNs because they can increase computational complexity and memory footprints. In an effort to further reduce computational complexity, there is interest in the use of linear transformations that maximize energy compaction yet yield real-valued spectra.

Naturally occurring signals are often non-periodic signals and are more amenable to being estimated as stationary first-order Markov processes, or, more generally, auto-regressive models of order one, AR(1). For this reason, we propose the use of the discrete cosine transform (DCT) over the DFT for efficient energy compaction in natural signals. In fact, the commonly used compression algorithm JPEG (Wallace, 1991) also makes use of the DCT for this reason.

The DCT uses real-valued cosine functions mapping an input sequence $\mathbf{x}$ from $\mathbb{R}^N \rightarrow \mathbb{R}^N$. For an AR(1) process, the DCT is the optimal KLT basis (Unser, 1984; Torun & Akansu, 2013) (also see appendix). This enables the DCT to express AR(1) processes in fewer components than the DFT. In turn, this increases the energy compaction properties of the transform, allowing for a higher decimation rate while preserving a similar amount of signal content.

Because the DCT (and DFT) operate on discrete signals, there must be a definition for the boundary conditions for both the left and right boundaries of the repeated sequence along with the point at which the function is defined as even. This gives rise to various definitions of the DCT—a total of $8$. In this work, we refer to the orthonormal DCT type-II as "the DCT." Boundary conditions (implied periodicity) for the DCT-II are even at $n = -\frac{1}{2}$ and $n = N - \frac{1}{2}$ and can be seen in Figure 1. The single dimension orthonormal DCT-II is given by Equation (1) and is straightforward to extend to multiple dimensions:

$$X_k = \sqrt{\frac{1}{N}}x_0 + \sum_{n=0}^{N-1} x_n \sqrt{\frac{2}{N}} \cos\left[\frac{\pi}{N}\left(n + \frac{1}{2}\right)k\right] \qquad \text{for } k = 0, \dots N-1. \qquad (1)$$

## 3 DCT-DIFFSTRIDE

Similarly to the architectures in (Rippel *et al.*, 2015; Riad *et al.*, 2022), DCT-DiffStride, depicted in Figure 1, utilizes a cropping mask on the frequency-domain representation of the input signal. This is then followed by the inverse transform on the cropped frequency signal to transform back into the spatial domains. For example, an input image $\mathbf{x} \in \mathbb{R}^{H \times W}$ is transformed by the DCT resulting in $\mathbf{X} \in \mathbb{R}^{H \times W}$. $\mathbf{X}$ is then cropped which is equivalent to decimation. Thus by adjusting the size of the crop, the stride of the filtering operation is also adjusted. Because the cropping operation itself is not differentiable, a stop gradient operator (Bengio *et al.*, 2013) is used so that the DCT-DiffStride layer is still differentiable with respect to the learnable decimation rates, $S$.

The output of the DCT is not necessarily symmetric, unlike the DFT for a real-valued input. Consequently, the learned cropping mask does not need to be symmetric to reconstruct the signal. To produce the windowed mask, we use a similar implementation to (Riad *et al.*, 2022) based on the adaptive attention span proposed by Sukhbaatar *et al.* (2019); however, we account for the differences in symmetry. DCT-DiffStride is able to produce a smaller minimum sized feature map than DiffStride because this symmetry does not need to be kept. The minimum output shape for a DCT-DiffStride layer is given by $1 + R$ and is $2 + 2R$ for DiffStride where $R$ is a defined smoothing factor for the cropping mask. Although this result may be inconsequential for high dimensional $\mathbf{x}$,

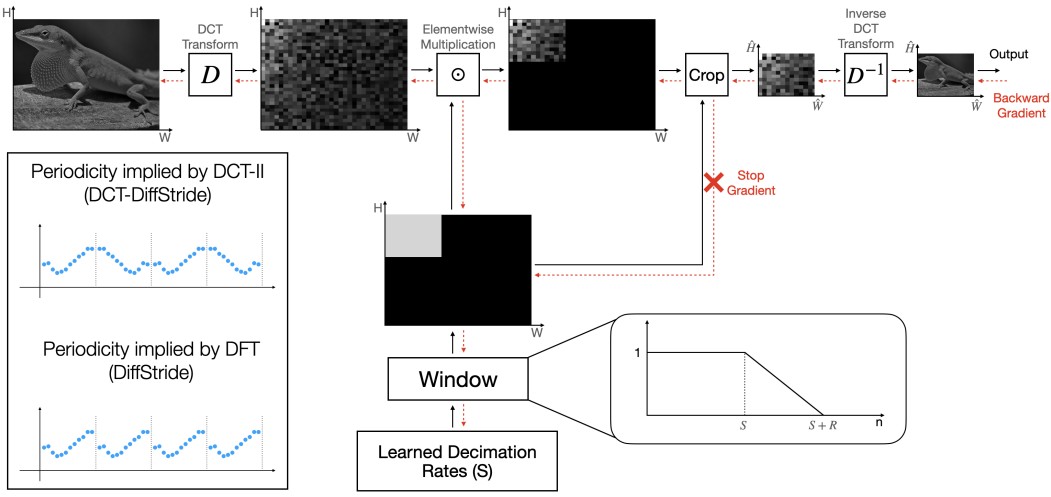

Figure 1: An overview of DCT-DiffStride on a greyscale image from ImageNet (Deng *et al.*, 2009).

the percentage of compression for lower-dimensional signals may be largely impacted. For example, if we use a smoothness of $R = 4$ with $\mathbf{x} \in \mathbb{R}^{32}$, as in many image datasets, DiffStride can compress to approximately 30% whereas DCT-DiffStride can compress to approximately 15% of the original size for a single layer.

### 3.1 REGULARIZATION

Riad *et al.* (2022) proposed a novel regularization method to promote better usage of time and memory complexity given by $\sum_{l=1}^{L} \prod_{i=1}^{l} \frac{1}{S_h^i \cdot S_w^i}$. There are a few drawbacks of this method. First, the model could learn to increase the striding factor therefore decreasing the regularization loss, but not reducing the feature map size. This is because of the max and min operations that keep the feature dimensionality from collapsing and retaining the smoothness factor. In other words, strides can be unconstrained and have the network decrease loss without any functional value. Second, although the regularization loss is proportional to model complexity, there is not a direct interpretation of the regularization value since the strides are unconstrained. While increasing the weight of the regularization term gives the network incentive to reduce memory and time complexity, the regularization term lacks semantic value. We propose a novel regularization term that alleviates these concerns by expressing model complexity as a percentage.

DCT-DiffStride reduces the feature map size along the spatial/temporal axes; however, it does not change the number of channels or the number of parameters in neighboring layers. As such, we define the model complexity, $C$, as a function of the spatial/temporal feature map dimensionality defined as $C = \sum_{l=1}^{L} \prod_{d=1}^{D} \frac{E_{l,d}}{M_{l,d}}$. $L$ is the number of convolutional layers, $D$ is the dimensionality of the spatial/temporal axes, and $E$ is the expanse (*e.g.*, height or width for an image). $M$ is the maximum possible expanse (*i.e.*, no decimation is performed in the network). We note that this does not include complexity of fully connected layers as they are also kept constant in the network.

The output shape of each DCT-DiffStride layer is a function of the learned stride for each particular dimension, the employed smoothness factor, and the minimum allowed feature map size. The minimum feature map size for DCT-DiffStride is 1 because the output of the DCT is not symmetric. For DiffStride, the minimum size is 2 to account for the symmetry in the DFT for real-valued, even-length signals. The output shape for a single dimension of a DCT-DiffStride layer is represented as $\lfloor \max(\frac{E_{l,d}}{S_{l,d}}, 1) \rfloor + R$ where $E$ is the expanse, $l$ represents the layer, $d$ is the dimension, $S$ is the learned stride, and $R$ is the smoothness factor. The output shape calculation for DiffStride is similar, but symmetry for real-valued inputs is taken into account giving $\lfloor \max(\frac{E_{l,d}}{S_{l,d}}, 2) \rfloor + 2R$. In our regularization term, we normalize each output shape by the maximum shape that the output could

be, $M$ (*i.e.*, no decimation is performed). This allows each regularization term to be interpreted as the percentage of total complexity for a given layer and dimension on a normalized range of [0, 1].

We propose a novel regularization method in this work with two variants that alleviate issues of interpretability and introduce implicit constraints keeping the loss bounded. In practice, there may be computational or memory limitations on the device that the model will be deployed. By specifying the maximum desired percentage model complexity $\hat{c}$, Equation (2) can be used to incentivize the network to not exceed the desired complexity while still allowing the network to learn the optimal rate of decimation for each layer. $L$ is the total number of DCT-DiffStride layers. This allows the network to learn a less complex model than $\hat{c}$ (*i.e.*, learn larger decimation rates) and optimize other loss terms such as cross entropy. In traditional decimation methods, the size of each layer would be designed apriori to keep the model within $\hat{c}$ complexity, but the proposed method allows the network to learn a smaller model while determining which layers should have higher/lower decimation rates through gradient descent. This method can also be generalized to specify percentage complexity to specific layers if overall model complexity is not sufficient.

$$\mathcal{L}_{\text{max complexity}} = \max\left(C - \hat{c} * L, 0.0\right) \qquad (2)$$

While Equation (2) is likely the method to employ in real-world situations, this work aims to illustrate an advantageous tradeoff in model complexity and model performance for the use of DCT-DiffStride against DiffStride. In order to fairly compare the two methods, each method needs to be evaluated with similar model complexity values. We propose a slight modification to Equation (2) such that the models will have very similar complexity values seen in Equation (3). Here, the network is penalized for having a larger or smaller overall complexity than $\hat{c}$ rather than just creating an upper bound seen in Equation (2). It is notable that the network is still able to learn decimation rates for each DCT-DiffStride layer, but the overall model complexity needs to match a value close to $\hat{c}$. This is purely to illustrate the tradeoff in model complexity and model performance.

$$\mathcal{L}_{\text{sq complexity}} = (C - \hat{c} * L)^2 \qquad (3)$$

For the remainder of this work, we use Equation (3) to ensure a fair comparison between DCT-DiffStride and DiffStride. The overall loss function for all experiments in this work is $\mathcal{L} = \mathcal{L}_{\text{cross entropy}} + \lambda \mathcal{L}_{\text{sq complexity}}$. We set $\lambda = 100$ for all experiments to make the regularization the dominant term in the overall loss calculation. This helps to ensure both DCT-DiffStride and Diffstride have a similar model complexity for ease of comparison.

## 4 EXPERIMENTS

We compare DCT-DiffStride and DiffStride on five classification tasks. We include both image and audio datasets that include naturally occurring signals, as well as a modulation classification dataset representing non-natural signals. Providing a wide variety of dataset types allows exploration into advantages that DCT-DiffStride and DiffStride may possess as well as expanding the the general number of applications for both methods. To ensure stability during training, all methods utilize unit $\ell^2$-norm global gradient clipping. In this work, we apply DCT-DiffStride between convolutional layers, but it can be applied between other architectural layers (*e.g.*, transformer layers) with minimal modification.

While Riad *et al.* (2022) showed that DiffStride is resilient to different stride initializations, our work aims to illustrate the tradeoff for DCT-DiffStride in terms of model complexity and model performance when compared to DiffStride. All models are trained with categorical cross-entropy loss along with the appropriate regularization penalty for each model complexity, $\hat{c}$.

### 4.1 IMAGE DATA

**Experimental Setup, CIFAR**  We compare DCT-DiffStride and DiffStride on various image classification tasks including CIFAR10, CIFAR100, and ImageNet. Each comparison uses the ResNet18 (He *et al.*, 2016) architecture with a residual block using the same definition as (Riad *et al.*, 2022). Each image experiment applies *mixup* (Zhang *et al.*, 2018) with $\alpha = 0.2$, which regularizes the

CNN using convex combinations of training instances. Convolutional layers use a weight decay of $5e-3$, and decimation rates are initialized to $S = [1, 2, 2, 2]$.

Due to their smaller size, we first benchmark DCT-DiffStride on the CIFAR datasets (Krizhevsky *et al.*, 2009) with an increased number of $\hat{c}$ values in order to better characterize the complexity tradeoff. For each CIFAR experiment, we utilize stochastic gradient descent (SGD) (Saad, 1999) with an initial learning rate of $0.1$, momentum of $0.99$ (Qian, 1999), and a batch size of 32. We train the models for a total of 80 epochs with a learning rate decay factor of $0.1$ at epochs 20, 40, and 60 with a final learning rate of $1e-4$. We use $\hat{c} = [0.1, 0.2, 0.3, ..., 0.8]$ to develop a tradeoff curve in terms of model complexity and model performance.

We use the same model definition as (Riad *et al.*, 2022), and we report the highest performing model from their work using traditional strided convolutions to establish a baseline to compare to traditional methods.

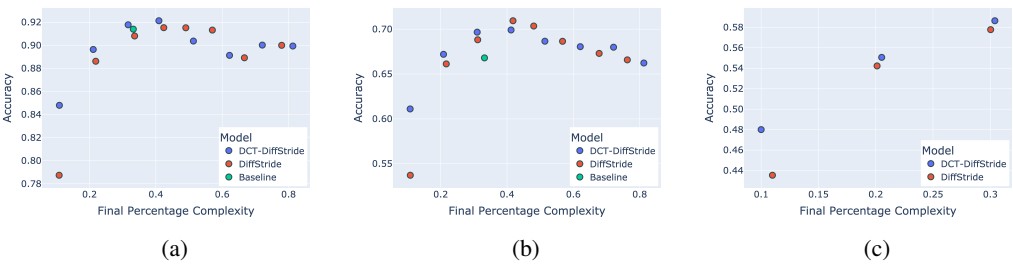

(a)             (b)             (c)

Figure 2: Accuracy over various model complexities with (a) showing CIFAR10, (b) showing CIFAR100 results, and (c) showing ImageNet results.

**Results, CIFAR**     Figure 2 shows the results for both CIFAR datasets, showing top-1 accuracy versus the model complexity. As expected, results are similar as CIFAR100 uses the same images as CIFAR10, but with more detailed labeling. As model complexity decreases, DCT-DiffStride outperforms DiffStride. The smaller the complexity, the larger the gap in performance indicating that DCT-DiffStride is able to utilize a higher decimation rate than DiffStride. For resource-constrained devices, this result would enable higher model performance without the need of upgrading memory capacity. Additionally, DCT-DiffStride obtained the best performing model with model complexity of approximately 40%. Because DCT-DiffStride and DiffStride both utilize low-pass filters to reduce complexity, this result could indicate the DCT needs fewer coefficients in the lower frequencies to represent image signals, compared to the DFT. Because the CIFAR datasets include many examples of naturally occurring images, this supports our hypothesis that the DCT's energy compaction property can better represent the energies in natural images. DCT-DiffStride also outperforms the baseline method with similar model complexities. This indicates that DCT-DiffStride is able to learn more optimal fractional strides at different layers.

**Experimental Setup, ImageNet**     We perform similar experiments on the ImageNet dataset (Deng *et al.*, 2009) containing 1,000 classes. We use the official training split with 1.28M training images and report results on the 50K validation images (50 images per class). We use the $64 \times 64$ resized dataset provided by Chrabaszcz *et al.* (2017) to ensure consistent image sizes. On ImageNet, we keep the same training procedure as the CIFAR datasets, but train the models for 45 epochs with an initial learning rate of $0.1$ with a learning rate decay factor of $0.1$ at 15 epoch intervals. Observing a large gap in performance between DCT-DiffStride and DiffStride at low model complexities, we use $\hat{c} = [0.1, 0.2, 0.3]$ for ImageNet to see if a similar result occurs on a larger dataset as well as a dataset consisting of larger sized images.

**Results, ImageNet**     A similar result was found on ImageNet as the CIFAR datasets. DCT-DiffStride achieved a much higher evaluation accuracy of 48% compared to that of 43.5% using DiffStride at 10% model complexity, and the gap decreases as more intermediate activations are retained. This reaffirms that DCT-Diffstride is able to use larger decimation rates than DiffStride even on larger images, while maintaining strong performance.

## 4.2 AUDIO DATA

**Experimental Setup**    In addition to image data, we evaluate DCT-DiffStride on the speech commands dataset (Warden, 2018). Raw audio data is sampled at 16KHz and processed into 64-dimensional log mel-scaled spectrograms using a window size of 25ms and 10ms overlap (Stevens *et al.*, 1937). All spectrograms are mean-normalized for each mel bin. In order to leverage batch training, audio is randomly cropped or padded to 1s in length. We use the adaptive momentum (AdaM) (Kingma & Ba, 2015) optimizer with an initial learning rate of 0.1 and a decay factor of 0.1 at epochs 20 and 40 with a batch size of 32 for a total of 60 epochs. To establish a baseline to compare against traditional methods, we report the accuracy from (Riad *et al.*, 2022) using traditional strided convolutions.

Similarly to (Riad *et al.*, 2022), a 2D-CNN based on (Tagliasacchi *et al.*, 2019) is employed where convolutional blocks consist of three convolutions. First, a $(3 \times 1)$ kernel is applied over time followed by a $(1 \times 3)$ kernel over frequency and a residual convolution. These convolutional blocks are illustrated in Figure 3a. The architecture consists of 5 convolutional blocks each followed by a decimation layer. The architecture then employs global average pooling and a single linear layer to perform classification. A decimation layer is also performed on the input to the network. The number of channels for each convolutional block are $[128, 256, 256, 512, 512]$ and decimation rates are initialized to $S = [2, 2, 1, 2, 1, 2]$.

**Results**    The accuracy versus model complexity is shown in Figure 3b. While both DCT-DiffStride and DiffStride perform comparably, DiffStride was found to have a slight advantage over DCT-DiffStride for the speech commands dataset. Both methods were able to use large decimation rates without loss in performance indicating redundant activation values. Performance was found to decrease at high model complexity values. Generalization performance may decrease without a large amount of decimation possibly due to spurious activations influencing predictions. Both methods outperformed the baseline method at a similar model complexity further demonstrating the ability for the network to learn superior decimation rates.

A possible explanation for the similar performances is that speech is comprised of many near periodic (*e.g.*, vowel sounds and semivowels) and non-periodic sources (*e.g.,* voiceless fricatives and plosives). Thus, no clear advantages in periodicity or boundary conditions exists between the DCT and DFT. Moreover, the energy concentrations in speech are more dispersed among phonemes, which may explain why the energy compaction of the DCT does not provide a distinct advantage.

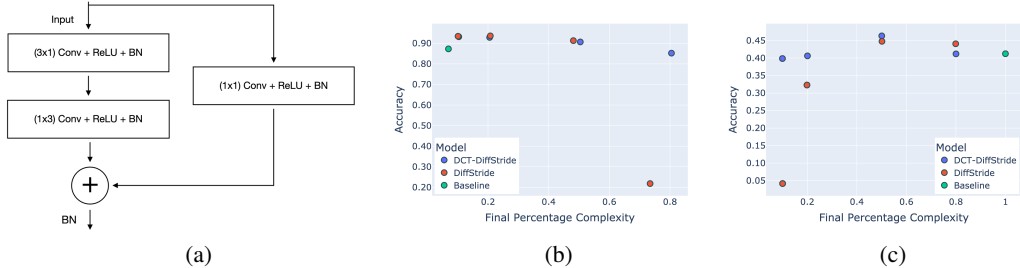

|         (a)          |         (b)          |         (c)          |

Figure 3: Parts (a) and (b) illustrate the speech commands experiment with (a) showing a convolutional block for audio data where *BN* stands for batch normalization and (b) showing top-1 accuracy over various model complexities for the speech commands dataset. (c) demonstrates accuracy over various model complexities for the RadioML 2018.01A dataset for signals in $[-14, 14]$dB SNR.

## 4.3 NON-NATURAL SIGNALS

Due to low-pass filtering in Fourier-based decimation methods such as DiffStride and spectral pooling, these methods are traditionally applied to natural signals as previously explained. However, low-pass frequency-based decimation methods can be applied outside the domain of natural signals. For example, even though signals may contain a large amount of high frequency content, successive non-linear transformations may be able to project the signal into a latent space that can more easily be represented by low frequency content.

**Experimental Setup**    To investigate this, we evaluate DCT-DiffStride on the RadioML 2018.01A (O'Shea *et al.*, 2018) dataset with the task of modulation scheme classification. There are 24 different classes with a total of 2.56M signals $S(T)$, each represented as a 2-dimensional vector consisting of in-phase (I) and quadrature (Q) components where $S(T) = I(T) + jQ(T)$. Observations range from $-20$dB to $+30$dB signal to noise ratio (SNR) in 2dB increments for a total of 26 different signal-to-noise ratio (SNR) values. The dataset is balanced across modulation type and SNR value with a total of 4,096 observations per {modulation type, SNR} pair. The 24 candidate modulation schemes fall under the following representative groups: (1) Amplitude, {*OOK, 4/8ASK, AM-SSB-SC, AM-SSB-WC, AM-DSB-WC, AM-DSB-SC*} (2) Phase, {*BPSK, QPSK, 8/16/32PSK, OQPSK*} (3) Amplitude and Phase, {*16/32/64/128APSK, 16/32/64/128/256QAM*} and (4) Frequency, {*FM, GMSK*}.

While there is not an official training split, we follow a similar procedure as in (O'Shea *et al.*, 2018; Harper *et al.*, 2020; 2021) where 1M observations are randomly selected for training and 1.56M observations are chosen for evaluation. Observed in (O'Shea *et al.*, 2018; Harper *et al.*, 2020; 2021), classification performance plateaus approximately below $-14$dB and above $14$dB SNR. In this work, we subset the training and evaluation datasets to only include signals in the range [-14, 14]dB SNR resulting in approximately 575K training observations and 900K evaluation observations.

We utilize a 1D CNN convolving over the temporal axis with 1D DCT-DiffStride and DiffStride layers. We base our model on (Harper *et al.*, 2020) utilizing mean and variance pooling after convolutional layers, a common technique used by x-vector architectures (Snyder *et al.*, 2018). We train a baseline model which uses an equivalent architecture to the model in (Harper *et al.*, 2020), but trained on signals in the specified dB range. To compare DCT-DiffStride and DiffStride, the decimation methods are applied following each convolutional layer. We use the adaptive momentum (AdaM) (Kingma & Ba, 2015) optimizer with an initial learning rate of $0.01$ and a decay factor of $0.1$ at epochs 15, 30, and 45 with a batch size of 128 for a total of 60 epochs. All decimation rates are initialized to one.

Table 1: RadioML 2018.01A results for each modulation category. Best performing models for each model complexity and modulation grouping are shown in bold. To make the table more compact, *DCT* and *DFT* represent DCT-DiffStride and DiffStride respectively.

| Modulation Group | $\hat{c} = 0.1$ | | $\hat{c} = 0.2$ | | $\hat{c} = 0.5$ | | $\hat{c} = 0.8$ | | Baseline |
| --- | --- | --- | --- | --- | --- | --- | --- | --- | --- |
| | DCT | DFT | DCT | DFT | DCT | DFT | DCT | DFT | |
| Frequency | **0.75** | 0.00 | **0.75** | 0.67 | **0.74** | 0.65 | 0.65 | **0.67** | 0.62 |
| Amplitude | **0.57** | 0.00 | **0.56** | 0.45 | **0.61** | 0.59 | 0.57 | **0.58** | 0.56 |
| Phase | **0.38** | 0.00 | **0.40** | 0.31 | **0.51** | 0.40 | 0.39 | **0.44** | 0.36 |
| Amplitude and Phase | **0.19** | 0.11 | **0.22** | 0.16 | 0.26 | **0.32** | 0.25 | **0.28** | 0.29 |

**Results**    Although modulation data is non-natural, Figure 3c and Table 1 clearly illustrate that DCT-DiffStride and DiffStride are able to classify modulation schemes even at low model complexities. Interestingly, both architectures outperformed the baseline architecture that does not use decimation. DCT-DiffStride was found to outperform DiffStride when the decimation rate was increased (*i.e.*, lower model complexity)—consistent with our image dataset results. DCT-DiffStride produced the best overall performing model with a model complexity of approximately 50%.

It is interesting that DCT-DiffStride and DiffStride are able to perform well with modulation data, particularly at small model complexities, because it is well known that modulation data contains a number of high frequency components that are important for classification. One possible explanation for this relates to the concept of *neural collapse* (Papyan *et al.*, 2020; Han *et al.*, 2022; Zhang *et al.*, 2021; Belkin *et al.*, 2019), which has shown that models trained with categorical cross-entropy tend to produce activations very close to the mean class value in later layers. Since in spectral analysis the zeroth entry in the DFT or DCT output is equivalent to the mean value of the signal, it may be that DCT-DiffStride and DiffStride can leverage neural collapse. More specifically, as networks are trained the within-class variability decreases under neural collapse which, in turn, decreases the

high frequency content in later layer activations and increases low frequency content. Because DCT-DiffStride and DiffStride operate as low-pass filters, they may be able to exploit this phenomenon.

## 5 DECIMATION RATE AND AR(1) RELATIONSHIP

In this section, we investigate the relationship between how well AR(1) processes fit the given datasets and intermediate activations along with learned decimation values. Because the DCT is the optimal KLT basis for an AR(1) process (see appendix), we would expect the trained networks to have larger decimation rates when the intermediate activations for a given layer are better modeled by an AR(1) process for DCT-DiffStride. To investigate this hypothesis, we take 100 random samples from each dataset, compute the activations for each layer for the 10% complexity model, fit an AR(1) process for each channel, and store the absolute value of the residuals between the activations and the AR(1) predicted activations for each sample and layer. We use the *statsmodels* package (Seabold & Perktold, 2010) to perform the AR(1) fitting. For 2-D data, the appendix describes the unraveling process to unpack the data into a 1-D series that can be modeled by an AR(1) process. The distribution of absolute residuals can be seen in the box plots and the feature map shape following decimation for the given layer are shown in the dotted lines in Figure 4. Feature map shape is defined as height×width for 2-D data and length for 1-D data (RadioML 2018.01A dataset). For the speech commands dataset, decimation is performed directly on the input whereas decimation is not performed until after convolutional layers in the remaining datasets. For this reason, the residual box plots for the input and the first layer are the same for the speech commands dataset.

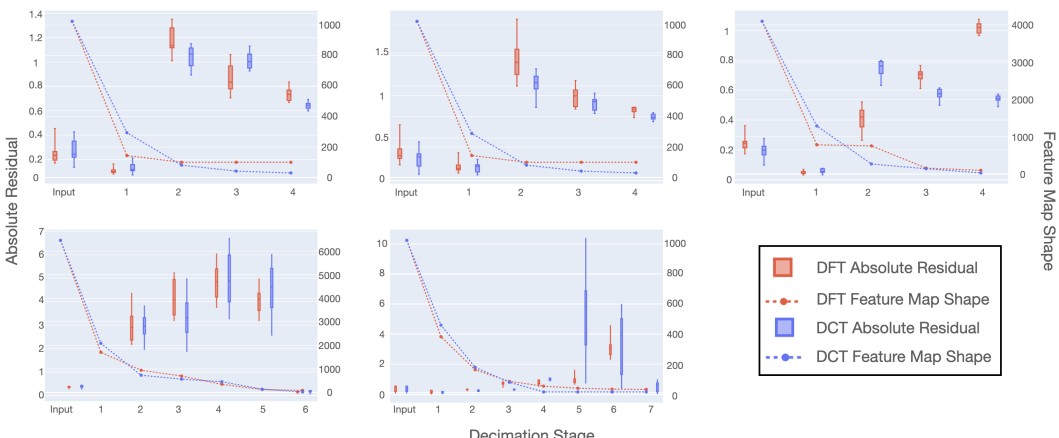

Figure 4: Absolute residuals (box plots) and learned decimation shapes for each layer. The top row, left to right: CIFAR10, CIFAR100, and ImageNet. The second row, left to right: speech commands and RadioML 2018.01A.

A consistent trend was seen across the datasets for DCT-DiffStride. When the residuals were small for successive layers, the feature map was more aggressively decimated. When the residuals increased, decimation rate stabilized. This supports the hypothesis that DCT-DiffStride leverages the DCT for larger decimation rates when the data better fits an AR(1) process. This behavior provides further evidence as to why DCT-DiffStride outperforms DiffStride, particularly for low-complexity models.

## 6 CONCLUSION

We propose DCT-DiffStride, a method to perform differentiable decimation leveraging the energy-compaction properties of the discrete cosine transform. DCT-DiffStride is able to outperform competitors on various classification tasks, particularly when large decimation rates are used. While low-pass filter methods are traditionally applied to natural signals, we show that DCT-DiffStride generalizes outside this domain and can be used as a direct replacement to other decimation methods, such as DiffStride and max pooling, in more general applications.

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

# A  APPENDIX

## A.1  AR(1) FOR 2-DIMENSIONAL DATA

In order to fit 2-D data to an AR(1) process, we must have a method to transform the 2-D data to a 1-D series. However, the method chosen must not introduce edge effects that inhibit the ability for an AR(1) process to fit the data. For example, neighboring pixels in an image are typically correlated with one another. Unpacking the 2-D data by flattening the rows where the end of the preceding row is adjacent to the start of the following row could introduce edge effects as the pixels could be very far from one another and be less likely to be correlated and fit an AR(1) process. Instead, we unpack 2-D data by traversing the rows ensuring unpacked data points remain close to points from the original 2-D structure. This procedure can be seen in Figure 5.

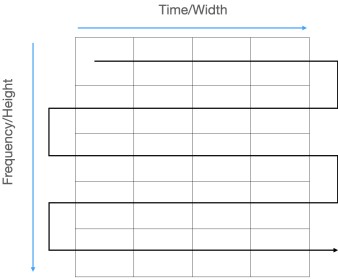

Figure 5: Overview of unpacking a 2-D sample into a 1-D series in order to fit an AR(1) process.

## A.2 Linear Transformation to Maximize Energy Compaction in Independent Spectral Coefficients: The KLT Transform

In our work, we make use of the DCT to leverage its energy-compaction properties to enable learnable decimation rates. In this section, we provide the mathematical foundation explaining the DCT's ability to represent AR(1) processes with fewer terms than the DFT enabling larger decimation rates without compromising classification performance.

It is desirable to use a transformation that tends to compact a large fraction of a signal's energy into just a few transform (or "spectral") coefficients. This characteristic is coupled with another desirable feature wherein maximum decorrelation is present among the transform coefficients, that is, the coefficients contain independent portions of the signal's energy components. Let us first define an $N$-sampled real-valued discrete signal as components of the vector, $\mathbf{x}$. The components of $\mathbf{x}$ are $x_k \in \mathbb{R}$ where $0 \leq k \leq N - 1$, or more succinctly, $\mathbf{x} \in \mathbb{R}^N$. For simplicity while attempting to be as general as possible, we assume that $\mathbf{x}$ is of the form of a normalized zero-mean random vector thus $E\{\mathbf{x}\} = 0$ and the $x_k$ are variates of a random variable, $X$, with zero-mean.

The covariance matrix of $\mathbf{x}$ with itself is its $N \times N$ autocorrelation matrix, $\mathbf{R_x} = Cov\{\mathbf{xx}\} = E\{\mathbf{xx}^T\}$. Note that the total energy of the signal, $\mathbf{x}$, assumed to be $||\mathbf{x}||^2$, is contained within the coefficients of matrix $E\{\mathbf{xx}^T\} = \mathbf{R_x}$. The component of $\mathbf{R_x}$ at row $k$ and column $j$ is denoted as $R_x(k, j)$ and encodes the correlation between the two signal components $x_k$ and $x_j$ comprising the energy term $|x_k x_j|$. Moreover, since $\mathbf{x} \in \mathbb{R}^N$, $\mathbf{R_x} = \mathbf{R_x}^T$.

**Observation 1:** $\mathbf{R_x}$ is a diagonal matrix.

If components $x_k$ and $x_j$ are uncorrelated, then $R_x(k, j) = 0$ for all $k \neq j$. Due to the definition of the autocorrelation matrix, it takes the form of a diagonal matrix with positive diagonal values equal to $|x_k|^2$. Because the signal $\mathbf{x}$ is of a form wherein the energy of each component, $x_k$, is proportional to $|x_k|^2$, the total energy of the signal is then the inner product of $\mathbf{x}$ with itself, or its squared $L_2$-norm, $||\mathbf{x}||^2$:

$$||\mathbf{x}||^2 = E\{\mathbf{x}^T \mathbf{x}\} = \sum_{k=0}^{N-1} R_x(k, k) \tag{4}$$

We denote the complex conjugate transpose of a matrix $\mathbf{A}$ as $\mathbf{A}^*$. Likewise, a conjugate transpose of a column vector $\mathbf{v}$ as the row vector $\mathbf{v}^*$. If the vector $\mathbf{v}$ is real-valued, as is the case for signal vector $\mathbf{x}$, then $\mathbf{v}^* = \mathbf{v}^T$. Let us apply a linear transformation, $\mathbf{T}$, to $\mathbf{x}$ as $\mathbf{y} = \mathbf{Tx}$ where $\mathbf{T}$ is an orthogonal transformation matrix. Since $\mathbf{T}$ is orthogonal, energy components in the spectral coefficients comprising $\mathbf{y}$ are independent. The physical principal of conservation of energy applies such that $||\mathbf{y}||^2 = E\{\mathbf{y}^T \mathbf{y}\} = ||\mathbf{x}||^2$.

We also note that the covariance of $\mathbf{y}$ with itself is its autocorrelation matrix, $\mathbf{R_y}$, as given in Equation (5):

$$\mathbf{R_y} = Cov\{\mathbf{yy}\} = E\{\mathbf{yy}^T\} = E\{\mathbf{Tx}(\mathbf{Tx})^*\} = E\{\mathbf{Txx}^T \mathbf{T}^*\} = \mathbf{T} E\{\mathbf{xx}^T\} \mathbf{T}^* = \mathbf{T R_x T}^* \tag{5}$$

Equation (5) indicates that if it is desirable to compact energy in the fewest $M < N$ components of the transformation vector $\mathbf{y}$, then the transformation matrix $\mathbf{T}$ should be structured such that the energy of $\mathbf{y}$ is contained within the first $M$ coefficients and all $y_k = 0$ for $M < k \leq N - 1$. We define the $M$-dimensional vector $\mathbf{y_M}$ to the the first $M$ coeffients of $\mathbf{y} = \mathbf{Tx}$. If a suitable transormation matrix $\mathbf{T}$ is chosen such that maximum energy compaction results, then from Equation (4), we have Equation (6).

$$||\mathbf{x}||^2 = ||\mathbf{y_M}||^2 = \sum_{k=0}^{M-1} |y_k|^2 = E\{\mathbf{y_M}^T \mathbf{y_M}\} = E\{(\mathbf{Tx})^* \mathbf{Tx}\} = E\{\mathbf{x}^T \mathbf{T}^* \mathbf{Tx}\} \tag{6}$$

Because $\mathbf{y_M}^T = [y_0 \ y_1 \ ... \ y_{M-1} \ 0 \ ... \ 0] = \mathbf{x}^T \mathbf{T}^*$ and $\mathbf{y_M} = \mathbf{Tx}$, we see that $\mathbf{y} = \mathbf{Tx} = \mathbf{y_M}$ yet the $N - M$ row vectors of T are irrelevant since the $\mathbf{y_M}$ coefficients, $y_k = 0 \ \forall \ k > M$ when

the transformation $\mathbf{y} = \mathbf{Tx}$ is calculated. Thus, we replace the transformation matrix $\mathbf{T}$ with that of $\mathbf{T_M}$ where $\mathbf{T_M}$ has the same row vectors, $\mathbf{t}_k^*$, as matrix $\mathbf{T}$ for all $k < M$ and has zero-valued row vectors, $\mathbf{t}_k^* = \mathbf{0}^T$, for all $M < k < N - 1$, where $\mathbf{0}$ denotes the null vector. This implies that maximum energy compaction occurs when we rewrite the rightmost term of Equation (6) as given in Equation (7).

$$E\left\{\mathbf{x}^T \mathbf{T}^* \mathbf{Tx}\right\} = E\left\{\mathbf{x}^T \mathbf{T}_M^* \mathbf{T}_M \mathbf{x}\right\} = E\left\{\mathbf{x}^T \begin{bmatrix} \mathbf{t}_0 & \mathbf{t}_1 & \cdots & \mathbf{t}_M & \mathbf{0} & \cdots & \mathbf{0} \end{bmatrix} \begin{bmatrix} \mathbf{t}_0^* \\ \mathbf{t}_1^* \\ \vdots \\ \mathbf{t}_M^* \\ \mathbf{0}^T \\ \vdots \\ \mathbf{0}^T \end{bmatrix} \mathbf{x}\right\}$$

(7)

From Equation (4), we know that $||\mathbf{x}||^2 = ||\mathbf{y_M}||^2 = \sum_{k=0}^{N-1} R_x(k,k)$, and from Equation (6) that $||\mathbf{x}||^2 = ||\mathbf{y_M}||^2 = E\{\mathbf{x}^T \mathbf{T}^* \mathbf{Tx}\}$; therefore, only the diagonal values of $\mathbf{TT}^*$ result in non-zero coefficients. Alternatively, since $\mathbf{T}$ by definition is an orthogonal transform, the inner product of any two column vectors, denoted as $(\mathbf{t}_k, \mathbf{t}_j) = 0$ for all $k \neq j$. Thus, we can rewrite Equation (7) as

$$E\left\{\mathbf{x}^T \mathbf{T}^* \mathbf{Tx}\right\} = E\left\{\mathbf{x}^T \left(\sum_{k=0}^{M-1} \mathbf{t}_k^* \mathbf{t}_k\right) \mathbf{x}\right\}$$

(8)

Using matrix algebra identities, we re-arrange the terms of Equation (8) resulting in Equation (9).

$$E\{\mathbf{x}^T \mathbf{T}^* \mathbf{Tx}\} = E\{\mathbf{x}^T \sum_{k=0}^{M-1} \mathbf{t}_k^* \mathbf{x} \mathbf{x}^T \mathbf{t}_k\} = E\{\mathbf{x}^T \sum_{k=0}^{M-1} \mathbf{t}_k^* \mathbf{R_x} \mathbf{t}_k\}$$

(9)

Equating $||\mathbf{x}||^2 = ||\mathbf{y_M}||^2 = E\{\mathbf{x}^T \mathbf{T}^* \mathbf{Tx}\} = E\{\mathbf{x}^T \sum_{k=0}^{M-1} \mathbf{t}_k^* \mathbf{R_x} \mathbf{t}_k\}$ from Equations (6) and (8), and $||\mathbf{x}||^2 = E\{\mathbf{x}^T \mathbf{x}\} = \sum_{k=0}^{N-1} R_x(k,k)$ from Equation (4), we obtain Equation (10).

$$E\{\mathbf{x}^T \sum_{k=0}^{M-1} \mathbf{t}_k^* \mathbf{x} \mathbf{x}^T \mathbf{t}_k\} = \sum_{k=0}^{N-1} R_x(k,k)$$

(10)

Because the orthogonal eigenvectors, $\mathbf{t}_k$, serve as column vectors of the transformation matrix, $\mathbf{T}$, and $\mathbf{R_x}$ is a diagonal matrix, the spectral decomposition theorem applies to Equation (10) indicating that the expression $\mathbf{T}^* \mathbf{R_x} \mathbf{T}$ satisfies the eigendecomposition where the $R_x(k,k)$ are the $k^{th}$ eigenvalue, $\lambda_k$, for the eigenvector $\mathbf{t}_k$ as given in Equation (11).

$$R_x(k,k)\mathbf{t}_k = \lambda_k \mathbf{t}_k$$

(11)

Therefore, the optimal transformation matrix in terms of compacting maximal signal energy components for a signal, $\mathbf{x}$, in the fewest spectral coefficients such that all components contain independent, or decorrelated, energy values is in fact the covariance (or autocorrelation) matrix of $\mathbf{x}$ defined as $\mathbf{R_x} = Cov\{\mathbf{x}\} = E\{\mathbf{xx}^T\}$. Therefore, $\mathbf{T} = \mathbf{R_x}$ and is an orthogonal transform.

The particular transformation matrix, $\mathbf{T} = \mathbf{R_x}$, that maximizes energy compaction within as few independent or decorrelated spectral coefficients as possible is not a new result and was formulated by Harold Hotelling in 1933 (Hotelling, 1933) for the purposes of obtaining Principal Components in a sequence of values for statistical analyses and is thus sometimes referred to as the "Hotelling transform," or, in more more contemporary times, as "Principal Components Analysis" (PCA). In later publications by Kari Karhunen and Michel Loève, this result was again simultaneously obtained and published by both researchers (Loève, 1945; 1955; Karhunen, 1946) as a series expansion

method for representing continuous random processes, thus, $\mathbf{T} = \mathbf{R_x}$ is sometimes referred to as the "Karhunen-Loève transform" or the "KLT."

In the contemporary era of data analytics, the use of PCA to obtain $\mathbf{T} = \mathbf{R_X}$ is often applied to data sequences represented as $\mathbf{x}$ to estimate the data vector in a reduced dimensional space as was originally proposed by Hotelling. Modern PCA is typically used to determine a reduced set of dominant basis vectors whose linear combinations closely approximate the data vector $\mathbf{x}$ thus allowing it, or at least a close approximation to it, to be processed in a lower dimensional space as described in (Fukunaga, 1993).

### A.3 PRACTICAL TRANSFORMS TO MAXIMIZE ENERGY COMPACTION WITH INDEPENDENT SPECTRAL COEFFICIENTS

As is apparent from the previous derivations, the property of requiring the spectral coefficients to be independent, in our case, of representing decorrelated signal energy components, implies that the desired transformation be orthogonal as is the case of the KLT. Unfortunately, computation of the KLT transformation matrix, the autocorrelation of the signal $\mathbf{x}$, is computationally intense. In general, an $N$-dimensional vector $\mathbf{x}$, requires $N^2$ scalar multiplications and $N^2$ scalar additions of two operands. Other classes of orthogonal transformations are independent with regard to properties of the signal and comprise a fixed transformation matrix.

The use of a fixed transformation matrix is desirable since it may be used repeatedly and independent of the particular structure of the signal. Furthermore, some classes of these transformation matrices have desirable structural properties such as being represented as factors of sparse matrices enabling efficient algorithms to be devised that greatly reduce the number of arithmetic operations required for the computation of the spectral or transformation vector. In particular, the discrete Fourier transform (DFT) has both the property of a transformation matrix with a fixed structure, and of being decomposable into a set of sparse matrix factors. This observation has led to the development of the so-called "Fast Fourier Transform" (FFT) (Cooley & Tukey, 1965) that is an efficient implementation of the DFT. Since the publication of the FFT algorithm, which some attribute to work as early as that of Carl Gauss, many alternative efficient algorithms have been developed. A summary of more modern and effect spectral transformation methods and algorithms is described in (Thornton *et al.*, 2012).

### A.4 KLT FOR REAL-VALUED PERIODIC PROCESS: THE DISCRETE FOURIER TRANSFORM

In terms of comprising independent energy components, the DFT is often used due to the Wiener-Khinchine theorem that states the power spectral density of a physical signal is equivalent to the Fourier transform of the signal's autocorrelation function (Ziemer & Trantor, 1985). For these reasons, the DFT or FFT is often used to compute energy components of a signal.

The DFT is optimal whenever the $\mathbf{x}$ vector is comprised of periodic signal. That is, $x_k = x_{k+m}$ for all values of $k$. This can be shown to be the case since the autocorrelation matrix, $\mathbf{R_X}$, has a circulant structure.

$$\mathbf{R_x} = E\left\{\mathbf{x}\mathbf{x}^T\right\} = \begin{bmatrix} r_0 & r_1 & \cdots & r_{N-1} \\ r_{N-1} & r_0 & \cdots & r_{N-2} \\ \cdots & \cdots & \ddots & \cdots \\ r_1 & r_2 & \cdots & r_0 \end{bmatrix} \tag{12}$$

The eigenvectors of $\mathbf{R_x}$ as given in Equation (11) can be determined and are found to consist of the Fourier basis vectors, $\mathbf{w}_N^k$, used to construct the DFT transformation matrix as shown in Equation (13).

$$R_x\left(k,k\right)\mathbf{w}_N^k = \lambda_k\mathbf{w}_N^k \quad , \quad w_N^k = e^{-\frac{i2\pi k}{N}} \quad , \quad i^2 = -1 \tag{13}$$

Thus, the DFT is a special case of the KLT for strictly periodic signals. As a signal loses its periodic structure, the DFT becomes less optimal in terms of energy compaction, but retains its ability to preserve independence or de-correlated energy components in the spectral vector due to the

Wiener-Khinchine theorem. For this reason, the DFT is often the first choice for determining energy components of a signal due to the widespread popularity of efficient, or "FFT," algorithms and the fact that many signals in communications systems are in the form of modulated carrier waves and thus have a good degree of periodicity.

The spectral coefficients comprising the DFT transform are complex, even when the transformed signal $\mathbf{x}$ is real-valued. In an effort to further reduce computational complexity, there is interest in the use of linear transformations that maximize energy compaction yet yield real-valued spectra.

### A.5 KLT FOR AR(1) PROCESS: THE DISCRETE COSINE TRANSFORM

If the signal $\mathbf{x}$ is comprised of components, $x_i$, that are related as $x_k = \rho x_{k-1} + z_k$, where $z_k$ is a variate (i.e., the $k^{th}$ outcome) of the Gaussian distributed random variable $Z$ with zero-mean, $\mu_{\mathbf{Z}} = 0$, and variance $\sigma_{\mathbf{Z}}^2$ and the constant correlation coefficient, $\rho$, satisfies $|\rho| < 1$. In this case, $\mathbf{x}$ is a stationary first-order Markov process, or, more generally and autoregressive model of order one, AR(1).

The autocorrelation matrix for $\mathbf{x}$ is $\mathbf{R_x} = Cov\{\mathbf{x}\} = E\{\mathbf{xx}^T\}$ and is given in Equation (14) as follows from Lai (1978).

$$
\mathbf{R_x} = \sigma_{\mathbf{Z}}^2
\begin{bmatrix}
\rho^0 & \rho^1 & \rho^2 & \rho^3 & \cdots & \rho^{N-1} \\
\rho^1 & \rho^0 & \rho^1 & \rho^2 & \cdots & \rho^{N-2} \\
\rho^2 & \rho^1 & \rho^0 & \rho^1 & \cdots & \rho^{N-3} \\
\rho^3 & \rho^2 & \rho^1 & \rho^0 & \cdots & \rho^{N-4} \\
\vdots & \vdots & \vdots & \vdots & \ddots & \vdots \\
\rho^{N-1} & \rho^{N-2} & \rho^{N-3} & \rho^{N-4} & \cdots & \rho^0
\end{bmatrix}
\tag{14}
$$

We wish to compute the eigenvectors of $\mathbf{R_x}$ in Equation (14) and make use of the fact that the eigenvectors of $\mathbf{R_x}$ and $\beta^2 \mathbf{R_x^{-1}}$ are equivalent wherein the scalar constant $\beta^2$ is defined as:

$$
\beta^2 = \frac{\rho^2}{1 + \rho^2}
\tag{15}
$$

We also define the scalar constant, $\alpha$ in Equation (16) as:

$$
\alpha = \frac{\rho}{1 + \rho^2}
\tag{16}
$$

The form of $\beta^2 \mathbf{R_x^{-1}}$ is determined as follows from Mallik (2001) and is that of the tridiagonal matrix as shown in Equation (17).

$$
\beta^2 \mathbf{R_x^{-1}} = \frac{1}{\sigma_{\mathbf{Z}}^2}
\begin{bmatrix}
1-\rho\alpha & -\alpha & 0 & \cdots & & & & & 0 \\
-\alpha & \rho^0 & -\alpha & 0 & \cdots & \cdots & \cdots & \cdots & \vdots \\
0 & -\alpha & \rho^0 & -\alpha & \cdots & \cdots & \cdots & \cdots & \vdots \\
\vdots & 0 & -\alpha & \rho^0 & 0 & \cdots & \cdots & \cdots & \vdots \\
\vdots & \vdots & 0 & \ddots & -\alpha & 0 & \cdots & \cdots & \vdots \\
\vdots & \vdots & \vdots & 0 & \ddots & \ddots & 0 & \cdots & \vdots \\
\vdots & \vdots & \vdots & \vdots & -\alpha & \rho^0 & -\alpha & 0 & \vdots \\
\vdots & \vdots & \vdots & \vdots & 0 & -\alpha & \rho^0 & -\alpha & \vdots \\
\vdots & \vdots & \vdots & \vdots & 0 & 0 & -\alpha & \rho^0 & 0 \\
\vdots & \vdots & \vdots & \vdots & 0 & 0 & 0 & -\alpha & -\alpha \\
0 & 0 & 0 & 0 & & & & & 1-\rho\alpha
\end{bmatrix}
\tag{17}
$$

When the correlation coefficient is close to unity,$\rho \approx 1$, and the variance is normalized to unity, $\sigma_{\mathbf{Z}}^2 = 1$, the matrix $\beta^2 \mathbf{R_x^{-1}}$ can be simplified to that of Equation (18).

$$\mathbf{T}_c = \begin{bmatrix} 1-\alpha & -\alpha & 0 & \cdots & & & & & & 0 \\ -\alpha & 1 & -\alpha & 0 & \cdots & \cdots & \cdots & \cdots & \vdots \\ 0 & -\alpha & 1 & -\alpha & \cdots & \cdots & \cdots & \cdots & \vdots \\ \vdots & 0 & -\alpha & 1 & 0 & \cdots & \cdots & \cdots & \vdots \\ \vdots & \vdots & 0 & \ddots & -\alpha & 0 & \cdots & \cdots & \vdots \\ \vdots & \vdots & \vdots & 0 & \ddots & \ddots & 0 & \cdots & \vdots \\ \vdots & \vdots & \vdots & \vdots & -\alpha & 1 & -\alpha & 0 & \vdots \\ \vdots & \vdots & \vdots & \vdots & 0 & -\alpha & 1 & -\alpha & \vdots \\ \vdots & \vdots & \vdots & \vdots & 0 & 0 & -\alpha & 1 & 0 \\ \vdots & \vdots & \vdots & \vdots & 0 & 0 & 0 & -\alpha & -\alpha \\ 0 & 0 & 0 & 0 & & & & & 1-\alpha \end{bmatrix} \tag{18}$$

The eigenvectors of $\mathbf{T}_c$ are determined as per Da Fonseca (2007), leading to the result:

$$\mathbf{T}_c \mathbf{t}_{ck} = \lambda_k \mathbf{t}_{ck} \tag{19}$$

Explicitly, these eigenvectors are of the form as provided in Equation (20).

$$\mathbf{t}_{ck}^T = \begin{bmatrix} \frac{1}{\sqrt{2}} & cos\left(\frac{(2m+1)k\pi}{2N}\right) & \cdots & cos\left((N-1)\pi\right) \end{bmatrix}, \forall k = 1, 2, \cdots (N-1); m = 1, 2, \cdots (N-1) \tag{20}$$

The coefficients of $\mathbf{t}_{ck}^T$ are recognized as the components of the "discrete cosine transform" (DCT) matrix, $\mathbf{T}_c$ as defined in (Ahmed *et al.*, 1974). It is noted that the coefficients of $\mathbf{T}_c$ are related to the DFT coefficients in Equation (13) as $Re\left\{w_N^k\right\}$. As previously mentioned, the fact that the DCT yields a real-valued spectrum is advantageous from a computational complexity viewpoint as compared to the complex-valued spectra arising from the DFT.

