# OpenReview forum: "DCT-DiffStride: Differentiable Strides with Real-Valued Data"
_ICLR.cc/2023/Conference — Submitted to ICLR 2023_

### Official Review · Reviewer_NRJF · 2022-10-23

**Confidence:** 3
**Correctness:** 3
**Technical Novelty And Significance:** 2
**Empirical Novelty And Significance:** 2
**Recommendation:** 5

**Clarity, Quality, Novelty And Reproducibility:**

The paper is reasonably clearly written, with a few more striking things that need fixing:
 - "However, in many applications the DFT can often be outperformed by the discrete cosine transform (DCT) (Ahmed et al., 1974) due to relaxed assumptions of periodicity and better energy compaction" - I believe this statement is imprecise; the assumptions are not necessarily relaxed. They are just more fit for the problem.
 - "Naturally occurring signals are often defined as signals that originate in nature..." - this sentence needs rewriting.
 - "We note that this does not include complexity of linear layers as they are also kept constant in the network"- do the authors mean fully connected layers (convolutional layers are linear too)?
 - "In our regularization term, we normalize each output shape by the maximum shape that the output could be (i.e., no decimation is performed)" - I believe this normalization is implied for C in Eqs 3 & 4. However, it is not present in Eq 2 and should probably be fixed.
 - "However, it is clear that the use of frequency-based decimation methods can be applied" - this sentence needs to be rewritten.

**Strength And Weaknesses:**

## Strengths
DCT-DS is a sound, well-motivated idea, and the proposed model complexity measure and regularization term based on it seem reasonable too. The authors perform a reasonable amount of experiments on a good variety of data types. The authors show that in the low model complexity domain, DCT-DS can significantly outperform DS.

## Weaknesses
 - While the idea is sound and reasonably well-executed, it is quite incremental. This is not a problem in itself, but it is unclear how beneficial the method is. This is because, with the exception of the modulation scheme classification task, the authors only show comparisons between DS and DCT-DS, so it is not possible to assess how well these methods perform compared to more traditional methods.
 - As I understand, the only difference between learning the cutoff frequency for a given model complexity for DS or DCT-DS and setting the stride size manually for standard convolutional layers is that during training, the model can allocate the size reduction more flexibly. Therefore, it would also be important to discuss how the  "distribution" of size reduction across different layers differs between DS, DCT-DS and the traditional methods.
 - The authors note, "To ensure stability during training, all methods utilize unit l2-norm global gradient clipping"  - it is not obvious to me what causes instability in the case of DS and DCT-DS. Is this a weakness of these methods, or is training unstable anyways?

## Questions
 - Why does the squared penalty term in Eq 4 ensure that the model complexities of DiffStride and DCT-DiffStride will be better matched compared to Eq 3? Is it merely the fact that it encourages that the model complexity exactly matches $\hat{c}$ rather than just making it an upper bound?
 - Could the authors comment on why DiffStride overtakes DCT-DiffStride in Figure 2? Furthermore, why does the accuracy decrease as the model complexity increases? Is this simply showing overfitting behaviour, or is there a different reason?

**Summary Of The Paper:**

The authors consider the problem of reducing the size of hidden layers in convolutional neural networks (CNN). Size reduction is usually achieved by setting an appropriate stride for the convolutional layers. The stride is a hyperparameter and is usually fine-tuned using cross-validation. An alternative approach is DiffStride (DS, [1]). DS is based on the observation that strided convolutions over the activations of a CNN can be viewed as a low-pass filter over the Fourier-transformed activations. The cutoff frequency of a low-pass filter is a continuous parameter that can be learnt using gradient descent. Thus, the original recipe for DS is to compute the activations' discrete Fourier transform (DFT), remove frequencies above the learnable cutoff frequency and inverse DFT the filtered frequencies.

The authors propose extending DS by using the discrete cosine transform (DCT) instead of the DFT, as they argue that its properties are better suited for the problem at hand. The authors call their method DCT-DiffStride (DCT-DS). Furthermore, the authors propose a regularisation term to ensure that the model does learn a useful cutoff frequency instead of just keeping all the activations.

The authors benchmark their approach on a couple of standard image and audio classification tasks and a less standard modulation scheme classification task on radio signals.

References
[1] Rachid Riad, Olivier Teboul, David Grangier, and Neil Zeghidour. Learning strides in convolutional neural networks. ICLR, 2022

**Summary Of The Review:**

Overall reasonable, though quite incremental contributions mainly lacking some empirical studies/comparisons. I will be happy to raise my score if the authors address my concerns.

---

> ### Author Response · Authors · 2022-11-19
> **Response to Reviewer NRJF**
>
> Thank you for your time and feedback.  We have made changes to the paper and
> hope the responses below help to alleviate concerns and increase the quality of
> our work.
>
> > "DCT-DS is a sound, well-motivated idea, and the proposed model complexity measure and regularization term based on it seem reasonable too."
>
> The regularization methods proposed are more interpretable and can be more easily implemented than prior works.
>
> > "While the idea is sound and reasonably well-executed, it is quite incremental. This is not a problem in itself, but it is unclear how beneficial the method is. This is because, with the exception of the modulation scheme classification task, the authors only show comparisons between DS and DCT-DS, so it is not possible to assess how well these methods perform compared to more traditional methods."
>
> To address this concern, we report results found in related works into the speech commands and CIFAR datasets.  We found that DCT-DiffStride outperforms baseline methods with the experimented hand-tuned strides.  DCT-DiffStride also outperforms DiffStride on a variety of tasks presented in our work at the complexity values investigated, particularly at lower complexity values.
>
> > "As I understand, the only difference between learning the cutoff frequency for a given model complexity for DS or DCT-DS and setting the stride size manually for standard convolutional layers is that during training, the model can allocate the size reduction more flexibly. Therefore, it would also be important to discuss how the "distribution" of size reduction across different layers differs between DS, DCT-DS and the traditional methods."
>
> We have added section 5 which demonstrates how intermediate activations fit an AR(1) process and the resulting feature map size at each layer.  We show the lowest complexity models for DCT-DiffStride and DiffStride for each dataset in these figures.  It was found that DCT-DiffStride was able to decimate at larger rates earlier when activations more fit an AR(1) process.  DiffStride learned similar, although different feature map sizes, but there was a significant drop in performance using DiffStride instead of DCT-DiffStride at low complexities indicating DCT-DiffStride is more well suited for the investigated problems.
>
> > "The authors note, "To ensure stability during training, all methods utilize unit l2-norm global gradient clipping" - it is not obvious to me what causes instability in the case of DS and DCT-DS. Is this a weakness of these methods, or is training unstable anyways?"
>
> We did not observe that DCT-DiffStride or DiffStride themselves caused instability during training for the chosen tasks.  Training was found to be unstable for even the baseline method in the modulation dataset (RadioML 2018.01A).  To keep consistency across all datasets, we incorporated gradient clipping for all tasks.
>
> > "Why does the squared penalty term in Eq 4 ensure that the model complexities of DiffStride and DCT-DiffStride will be better matched compared to Eq 3? Is it merely the fact that it encourages that the model complexity exactly matches $\hat{c}$ rather than just making it an upper bound?
>
> Correct.  We note that the equation numbers have changed in the updated manuscript.  Equation 3 is now equation 2, and equation 4 is now equation 3.  In this response, we will refer to the updated equation numbers.
>
> While equation 2 would likely be used in practice, we use equation 3 to ensure data points fall around the same complexity for DCT-DiffStride and DiffStride so that the two can be more directly compared at specified complexity values.  We have slightly adjusted the wording in the paper to make this more clear.
>
> > "Could the authors comment on why DiffStride overtakes DCT-DiffStride in Figure 2? Furthermore, why does the accuracy decrease as the model complexity increases? Is this simply showing overfitting behaviour, or is there a different reason?"
>
> One possible reason is that DiffStride represents more of the relevant signal energy than DCT-DiffStride at particular complexity values.  However, the same can be said in favor of DCT-DiffStride at other values.   At high complexity values, we expect the performance of the DCT and DFT to be similar.  Both the DCT and DFT are invertible transforms; therefore, when all DCT/DFT data points are kept, 100% of signal energy is explained. Once sufficiently many points are kept, the DCT and DFT will converge to a similar amount of explained original signal energy.  The reason to use the DCT is that energy can be compacted in fewer points reaching the “sufficient” threshold earlier in the network, resulting in the ability to have larger decimation rates without sacrificing performance typically.

---

> > ### Author Response · Authors · 2022-11-19
> > **Continued Response to Reviewer NRJF**
> >
> > > "However, in many applications the DFT can often be outperformed by the discrete cosine transform (DCT) (Ahmed et al., 1974) due to relaxed assumptions of periodicity and better energy compaction" - I believe this statement is imprecise; the assumptions are not necessarily relaxed. They are just more fit for the problem.
> >
> > We agree and have adjusted the paper to be more precise.
> >
> > > "In our regularization term, we normalize each output shape by the maximum shape that the output could be (i.e., no decimation is performed)" - I believe this normalization is implied for C in Eqs 3 & 4. However, it is not present in Eq 2 and should probably be fixed.
> >
> > Thank you for pointing this out.  We have adjusted this equation to reflect this change.  We note that this equation is now incorporated in-line and no longer has an equation number.
> >
> > > "Naturally occurring signals are often defined as signals that originate in nature..." - this sentence needs rewriting.
> >
> > >"However, it is clear that the use of frequency-based decimation methods can be applied" - this sentence needs to be rewritten.
> >
> > We have adjusted the sentences in the updated manuscript.

---

### Official Review · Reviewer_Kc3t · 2022-10-25

**Confidence:** 3
**Correctness:** 3
**Technical Novelty And Significance:** 2
**Empirical Novelty And Significance:** 2
**Recommendation:** 6

**Clarity, Quality, Novelty And Reproducibility:**

There are concerns about the novelty and quality of the paper. Overall the paper is clear.

**Strength And Weaknesses:**

## Strengths
1. DCT is shown to be better to represent signals compactly and the paper shows its benefits compared DFT-diffstride.
2. Overall the paper is clearly written.

## Weaknesses
1. The novelty is somewhat limited as it is a straightforward modification to DFT-diffstride. Even though, marginal improvements are shown in practice it is not clear if it is sufficient for a publication.
2. Experiments simply compare against DFT and it is not clear if this could outperform hand-tuned strides in standard architectures. Furthermore, how difficult is it to optimize for the decimation rates, it seems it involves differentiating through DCT. Please comment on this.

**Summary Of The Paper:**

The paper presents a DCT-based approach to learning strides to perform spatial pooling. The approach is an improvement over DFT-diffstride and shows benefits over it in a low-complexity regime.

**Summary Of The Review:**

The paper is a straightforward improvement over DFT-diffstride and marginal improvements are shown against it.

## Post-rebuttal
I acknowledge the authors' response. As noted by other reviewers, the contribution is incremental and not sure if it is sufficient for publication at ICLR. Therefore I'm keeping the original score of marginal-accept.

---

> ### Author Response · Authors · 2022-11-19
> **Response to Reviewer Kc3t**
>
> Thank you for your time and feedback.  We have made changes to the paper and
> hope the responses below help to alleviate concerns and increase the quality of
> our work.
>
> > "The novelty is somewhat limited as it is a straightforward modification to DFT-DiffStride. Even though, marginal improvements are shown in practice it is not clear if it is sufficient for a publication."
>
> While we note the modification from the DFT to the DCT is straightforward, the improvement at low complexity values is significant.  At high complexity values, we expect the performance of the DCT and DFT to be similar.  Both the DCT and DFT are invertible transforms; therefore, when all DCT/DFT data points are kept, 100% of signal energy is explained. Once sufficiently many points are kept, the DCT and DFT will converge to a similar amount of explained original signal energy.  We have also incorporated baseline performances from other work.  DCT-DiffStride was found to outperform baseline methods of similar model complexity.
>
> Additionally, we introduce new regularization methods that are more interpretable than prior work.  We also have updated the paper to include section 5 which investigates the relationship between learned decimation rates and AR(1) fitness.  We believe this contribution increases generalizable knowledge about intermediate activations in CNNs and connects the motivation for using the DCT over the DFT with empirical evidence.
>
> > "Experiments simply compare against DFT and it is not clear if this could outperform hand-tuned strides in standard architectures."
>
> We note that more baseline methods could be trained to find hand-tuned strides.  To address this concern, we report results found in related works into the speech commands and CIFAR datasets.  We found that DCT-DiffStride outperforms baseline methods with the experimented hand-tuned strides.  DCT-DiffStride also outperforms DiffStride on a variety of tasks presented in our work at the complexity values investigated, particularly at lower complexity values.
>
> > "Furthermore, how difficult is it to optimize for the decimation rates, it seems it involves differentiating through DCT. Please comment on this."
>
> Thank you for pointing this out.  Optimization does involve differentiating through the DCT.  In traditional models, decimation rates are fixed.  For this reason, the computational graph is static during training.  This is not the case for DCT-DiffStride (or DiffStride).  Because the decimation rates are learned and can change during the training process, the computational graph must be updated when decimation rates are updated resulting in a change in feature map size.  This does slow down initial training as the network learns how to decimate; however, in our experiments, decimation rates stabilize quickly.  Training speed drastically increases as the decimation rates begin to settle as the computational graph does not need to be updated.
>
> While DCT-DiffStride is not able to take advantage of a static computational graph, it does remove the need for performing large-scale hyper parameter tuning to find optimal decimation rates which can be very time consuming and require computational power that can train multiple models.  In our experiments, this tradeoff appears to be in the favor of DCT-DiffStride and DiffStride over hyper parameter search.

---

### Official Review · Reviewer_LwJS · 2022-10-29

**Confidence:** 3
**Correctness:** 3
**Technical Novelty And Significance:** 3
**Empirical Novelty And Significance:** 2
**Recommendation:** 5

**Clarity, Quality, Novelty And Reproducibility:**

The paper is easy to read. Improvement is acceptable, but experiments should improved especially on audio domain.

**Strength And Weaknesses:**

1. This paper is well-written and easy to understand. The structure is very clear.

2. The architectures of this work are similar to DiffStride. The main innovation is the use of DCT. Knowledge in signal process is utilized. The energy compaction properties of DCT are well-known and widely used in lossy compression.

3. In Figure 2, DCT-DiffStride achieves the best performing model at 40% model complexity. However, for CIFAR-100, the baseline DiffStride seems achieve better performance at similar model complexity (around 40%) than DCT-DiffStride. Moreover, for ImageNet dataset, is it possible to provide a comparison figure between DCT-DiffStride and DiffStride as Figure 2?

4. For audio dataset, DCT-DiffStride and DiffStride is comparable. In Figure 3 (b), It seems DiffStride even achieves slightly higher accuracy than DCT-DiffStride at the lowest model complexity. Moreover, why the performance of DiffStride considerably decreases at the highest model complexity (the red point between 0.6 and 0.8)?


**Summary Of The Paper:**

This work extends the baseline model DiffStride by replacing the discrete Fourier transform (DFT) with the discrete cosine transform (DCT), which leverages the energy compaction properties of DCT. Experiments are conducted on natural signals non-natural signals. The results in image classification and modulation classification demonstrate the advantageous tradeoff in model complexity and model performance for DCT-DiffStride against DiffStride. In speech classification, at low model complexity, these two methods are comparable


**Summary Of The Review:**

Overall, the idea of this paper makes sense to me, but I think the experiments results and the novelty of this paper may be marginally below the acceptance.

---

> ### Author Response · Authors · 2022-11-19
> **Response to Reviewer LwJS**
>
> Thank you for your time and feedback.  We have made changes to the paper and hope the responses below help to alleviate concerns and increase the quality of our work.
>
> > "The architectures of this work are similar to DiffStride. The main innovation is the use of DCT. Knowledge in signal process is utilized. The energy compaction properties of DCT are well-known and widely used in lossy compression."
>
> > "In Figure 2, DCT-DiffStride achieves the best performing model at 40% model complexity. However, for CIFAR-100, the baseline DiffStride seems achieve better performance at similar model complexity (around 40%) than DCT-DiffStride."
>
> This was an interesting result.  While CIFAR-10 and CIFAR-100 contain the same images, the labeling is more fine-grained with CIFAR-100.  One possible reason for the higher performance of DiffStride over DCT-DiffStride on CIFAR-100 is that the DFT better represents the signal at a 40% complexity for finer-grained labels than the DCT.  We note that more complexity values could be tested to better represent the tradeoff curve for model complexity and model performance.  We chose increments of 10% due to time constraints; however, smaller increments may be able to illuminate where peak performance occurs for each approach.  We leave this to future work.
>
> > "Moreover, for ImageNet dataset, is it possible to provide a comparison figure between DCT-DiffStride and DiffStride as Figure 2?"
>
> We have adjusted the paper to include more data points for the ImageNet dataset.  Unfortunately, we were unable to create a curve up to 80% complexity in the rebuttal period; however, we were able to show a curve up to 30% complexity.  We were not able to establish a hyper parameter searched baseline for ImageNet to compare against also due to time constraints, but we point out that this is a key advantage of DCT-DiffStride.  Allowing the network to learn decimation rates on its own removes the need for arduous hyper parameter search.
>
> > "For audio dataset, DCT-DiffStride and DiffStride is comparable. In Figure 3 (b), It seems DiffStride even achieves slightly higher accuracy than DCT-DiffStride at the lowest model complexity."
>
> A possible explanation for the similar performances is that speech consists of many near periodic and non-periodic sources. Thus, no clear advantages in periodicity or boundary conditions exists between the DCT and DFT. Moreover, the energy concentrations in speech are more dispersed among phonemes, which may explain why the energy compaction property of the DCT does not provide a distinct advantage. Both methods may also be able to learn a way of representing intermediate activations that is more amenable to the DCT or DFT respectively.  In our paper update, section 5, figure 4 shows that aside from the initial decimation stage, the activations become less modeled by an AR(1) process which in turn may favor the DFT slightly.
>
> > "Moreover, why the performance of DiffStride considerably decreases at the highest model complexity (the red point between 0.6 and 0.8)?"
>
> We expect model performance to decrease with very large complexity values due to potential loss in generalization performance.  We report results on the testing set, unseen during the training process.  The considerable drop in performance may be due to spurious activations influencing predictions and overfitting.  The training accuracy for this datapoint was 96.8% indicating an overfit model.

---

> > ### Author Response · Authors · 2022-11-19
> > **Continued Response to Reviewer LwJS**
> >
> > > "Overall, the idea of this paper makes sense to me, but I think the experiments results and the novelty of this paper may be marginally below the acceptance."
> >
> > Our work builds upon DiffStride; however, we introduce new components increasing the ability and understanding of differentiable strides.  The use of the DCT has key advantages over the DFT.  The most simple of which is operating strictly in the real-valued domain of which current machine learning architectures are optimized.  However, our larger contribution is the understanding as to why the DCT is better suited for machine learning tasks than the DFT.  The energy compaction properties of the DCT are well known for AR(1) signals making it commonly used in image compression.  As expected, DCT-DiffStride outperforms DiffStride at lower complexities for the tested image datasets.  Surprisingly, DCT-DiffStride outperformed DiffStride on the RadioML 2018.01A dataset which is not in the AR(1) domain.
> >
> > To better explain this phenomenon, we have added section 5 which demonstrates correlations between how well an AR(1) process fits intermediate activations and the learned decimation rates.
> >
> > Additionally, we introduce new regularization methods that are more interpretable than previous works.  This allows for more control when customizing architectures for specific tasks particularly when memory usage is critical.  By decoupling the strides from previous layers, regularization can be more easily applied to a specific layer.  This may be useful for a variety of architectures.  For example, auto-encoder architectures where the latent representation may need to be at most a certain size.  The regularization term can be applied directly to a specific layer without the need of introducing regularization on preceding layers.  This is more interpretable and simpler for developers to implement.

---

### Official Review · Reviewer_yZ5T · 2022-11-01

**Confidence:** 4
**Correctness:** 3
**Technical Novelty And Significance:** 2
**Empirical Novelty And Significance:** 2
**Recommendation:** 3

**Clarity, Quality, Novelty And Reproducibility:**

Clarity: the paper has almost 4 pages of "methods" yet no pseudo-code or implementation details are given to the reader, and there are rather a lot of high-level description of time-frequency representations which could be useful if it served the narrative and the reader's comprehension.

Novelty: minimal change to DiffStride (replacing a time-frequency representation by another one)

Reproducibility: No code, no pseudo-code and a lack of implementation details make it difficult to reimplement.

**Strength And Weaknesses:**

Strengths:
* DCT-DiffStride is significantly better than DiffStride in the low computational cost regime.

Weaknesses:
* The main weakness of the paper is the minimal technical novelty (replacing DFT by DCT in DiffStride), which is hidden behind a lot of verbosity.
* Section 2.2 is very verbose and could be summarized. For instance, the example given for non-periodic signals is artificial. Also, how are properties of the DCT on AR(1) processes relevant unless authors justify they are a good model for the natural signals considered? What would have needed more details is the last paragraph which vaguely describes the implementation details.
* Similarly, the appendix provides a generic description of KLT/DCT/DFT that seems unconnected to the narrative of the paper.
* The weight of the regularization (100) is not discussed, neither how it was cross-validated.
* The results show that DCT-DiffStride only improves in the very low complexity regime. As such, and given that the technical contribution is minimal, it is hard to consider this paper as a significant contribution on top of DiffStride.

**Summary Of The Paper:**

Authors improve DiffStride, a pooling layer with learnable strides, by replacing the Fourier transform with a DCT.

**Summary Of The Review:**

DCT-DiffStride applies a simple tweak to DiffStride. Simplicity is not a flaw but rather a quality when 1) it serves better performance 2) is clearly justified and explained. Given the inconsistent experimental results, and the lack of analysis of the high performance in the low-computation regime, it's hard to consider this paper worth being published in its current state.

---

> ### Author Response · Authors · 2022-11-19
> **Response to Reviewer yZ5T**
>
> Thank you for your time and feedback.  We have made changes to the paper and hope the responses below help to alleviate concerns and increase the quality of our work.
>
> > "The main weakness of the paper is the minimal technical novelty (replacing DFT by DCT
> in DiffStride), which is hidden behind a lot of verbosity."
>
> We note that the change from the DFT to the DCT itself is a small change; however, the aim and contributions of our work can be summarized into the following points:
> 1. Further generalizable knowledge about the learning process specifically enhancing the understanding as to why the DCT is a more suitable choice than the DFT for learnable decimation relating to autoregressive processes and optimal bases.  However, we grant that our original submission lacked a clear connection between the motivation and empirical results.  As such, we added section 5 to further understand why the DCT is more suitable.
> 2. Introduce more interpretable and usable regularization methods for learnable decimation methods explained in section 3.1.
> 3. Increase the application domain of low-pass filter-based learnable decimation methods.  Traditionally, low-pass filters are used on natural signals; however, in section 4.3, we illustrate that these methods are also suitable for communication signals which typically contain significant energy in higher frequencies and are not considered “natural” signals.  In the newly added section 5, we illustrate that DCT-DiffStride is more suitable when intermediate activations can be modeled as an AR(1) process.  To our knowledge, we expand the class of problems that these methods can be applied to.
>
> > "Section 2.2 is very verbose and could be summarized. For instance, the example given for non-periodic signals is artificial"
>
> We agree that this section can be compacted and have reflected this change in the paper revision.
>
> > "Also, how are properties of the DCT on AR(1) processes relevant unless authors justify they are a good model for the natural signals considered..."
>
> Thank you for making this point.  We address this problem in section 5 that we described above.  We connect both the datasets and intermediate activations to an AR(1) process and enhance the understanding for why the DCT should be used over the DFT.  We have also adjusted the appendix to better connect to understanding of the paper.  Section 2 was also adjusted so that the connection between the DCT, KLT, and AR(1) processes in terms of motivation is more clear.
>
> > "The weight of the regularization (100) is not discussed, neither how it was cross-validated."
>
> The choice of 100 was chosen to make the regularization term dominant in the loss function over cross-entropy so that models trained with the DCT and DFT end with similar complexity values to establish a tradeoff curve.  This term is quickly optimized toward 0 (typically within the 1st epoch) as the network is still allowed to determine the decimation rates for each layer while the overall complexity over all layers is the large regularization.  Cross-entropy becomes dominant once the network has achieved the desired complexity.  We grant that this choice can and should be further cross-validated and we leave this to future work.
>
> > "The results show that DCT-DiffStride only improves in the very low complexity regime"
>
> We note that the performance gains are most notable at lower complexities; however, we still believe this is a significant result for the following reasons.
> 1. DCT-DiffStride outperforms baseline methods of similar complexities on CIFAR-10, CIFAR-100, and speech commands as well as outperforming the baseline architecture for the RADIOML 2018.01A dataset.
> 2. In situations where edge devices with little memory are expected to perform computation/store network activations, DCT works significantly better than the DFT.
>
> > "the paper has almost 4 pages of "methods" yet no pseudo-code or implementation details"
>
> We note this shortcoming, and we plan to release the codebase on github following the double blind review process.
>
> > "Simplicity is not a flaw but rather a quality when 1) it serves better performance 2) is clearly justified and explained. Given the inconsistent experimental results, and the lack of analysis of the high performance in the low-computation regime, it's hard to consider this paper worth being published in its current state."
>
> Baseline performances and section 5 have been added to address these concerns.  We found that DCT-DiffStride can learn larger decimation rates when activations are better modeled by an AR(1) process.  Additionally, due to the low complexity constraint on the models and given that activations at certain layers seem to be modeled well by an AR(1) process, this is one possible explanation as to why DCT-DiffStride typically outperforms DiffStride in the low-computation regime.

---

### Decision · Program_Chairs · 2023-01-20

**Decision:**

Reject

**Justification For Why Not Higher Score:**

 The reviewers generally felt that the proposed contribution is quite incremental and not yet up to the bar of an ICLR publication.

**Justification For Why Not Lower Score:**

 NA

**Metareview: Summary, Strengths And Weaknesses:**

Thank you for your submission to ICLR.  The reviewers agree that while there do appear to be some benefits of the approach, the contributions here (which largely involves replacing the DFT, from past work on differentiable strides, with the DCT) are likely too incremental to warrant acceptance to ICLR in its current form.  Though the authors do importantly point out the larger scope of these contributions, ultimately the consensus feeling of the reviewers is that the higher-level arguments of the authors were not fully satisfactory when it came to justifying this change from a broader perspective.

**Summary Of Ac-Reviewer Meeting:**